

# Identification and expression analysis of cytokinin metabolic genes *IPTs*, *CYP735A* and *CKXs* in the biofuel plant *Jatropha curcas*

Li Cai[1,2], Lu Zhang[1,3], Qiantang Fu[1] and Zeng-Fu Xu[1]

[1] Key Laboratory of Tropical Plant Resources and Sustainable Use, Xishuangbanna Tropical Botanical Garden, Chinese Academy of Sciences, Menglun, Mengla, Yunnan, China
[2] College of Life Sciences, University of Chinese Academy of Sciences, Beijing, China
[3] National Engineering Research Center for Ornamental Horticulture, Flower Research Institute of Yunnan Academy of Agricultural Sciences, Kunming, Yunnan, China

Corresponding authors
Qiantang Fu, fuqiantang@xtbg.ac.cn
Zeng-Fu Xu, zfxu@xtbg.ac.cn

## ABSTRACT

The seed oil of *Jatropha curcas* is considered a potential bioenergy source that could replace fossil fuels. However, the seed yield of *Jatropha* is low and has yet to be improved. We previously reported that exogenous cytokinin treatment increased the seed yield of *Jatropha*. Cytokinin levels are directly regulated by isopentenyl transferase (IPT), cytochrome P450 monooxygenase, family 735, subfamily A (CYP735A), and cytokinin oxidase/dehydrogenase (CKX). In this study, we cloned six *IPT* genes, one *JcCYP735A* gene, and seven *JcCKX* genes. The expression patterns of these 14 genes in various organs were determined using real-time quantitative PCR. *JcIPT1* was primarily expressed in roots and seeds, *JcIPT2* was expressed in roots, apical meristems, and mature leaves, *JcIPT3* was expressed in stems and mature leaves, *JcIPT5* was expressed in roots and mature leaves, *JcIPT6* was expressed in seeds at 10 days after pollination, and *JcIPT9* was expressed in mature leaves. *JcCYP735A* was mainly expressed in roots, flower buds, and seeds. The seven *JcCKX* genes also showed different expression patterns in different organs of *Jatropha*. In addition, CK levels were detected in flower buds and seeds at different stages of development. The concentration of $N^6$-($\Delta^2$-isopentenyl)-adenine (iP), iP-riboside, and *trans*-zeatin (tZ) increased with flower development, and the concentration of iP decreased with seed development, while that of tZ increased. We further analyzed the function of *JcCYP735A* using the CRISPR-Cas9 system, and found that the concentrations of tZ and tZ-riboside decreased significantly in the *Jccyp735a* mutants, which showed severely retarded growth. These findings will be helpful for further studies of the functions of cytokinin metabolic genes and understanding the roles of cytokinins in *Jatropha* growth and development.

## INTRODUCTION

*Jatropha curcas* is a multipurpose tree that belongs to the Euphorbiaceae family. It can endure drought and adapt to barren land in tropical and subtropical regions. *Jatropha* is considered a promising biofuel plant due to the high oil content in its seeds (*Akashi, 2012*; *Francis, Edinger & Becker, 2005*; *Makkar & Becker, 2009*). However, the seed yield is very low, potentially because of the relatively low number of total flowers and/or the ratio of female to male flowers in *Jatropha* (*Kumar & Sharma, 2008*; *Kumar Tiwari, Kumar & Raheman, 2007*; *Rao et al., 2008*). Recently, several studies have reported that exogenous cytokinin (CK) treatment can significantly increase the total number of flowers per inflorescence, the female-to-male flower ratio, and the seed yield (*Fröschle, Horn & Spring, 2017*; *Pan & Xu, 2011*).

Cytokinins are important hormones in plants and participate in many biological processes, such as apical dominance (*Shimizu-Sato, Tanaka & Mori, 2009*; *Tanaka et al., 2006*), root proliferation (*Kudo, Kiba & Sakakibara, 2010*; *Werner et al., 2003*), reproductive development (*Ashikari et al., 2005*), and senescence (*Gan & Amasino, 1995*). Endogenous CKs containing $N^6$-($\Delta^2$-isopentenyl)-adenine (iP), *trans*-zeatin (tZ), *cis*-zeatin (cZ), dihydrozeatin (DZ), and their conjugates are known as isoprenoid CKs (*Mok & Mok, 2001*). The major derivatives are generally iP- and tZ-type CKs (*Sakakibara, 2006*).

Cytokinin biosynthesis and degradation pathways have been well studied in the past decade (Fig. 1) (*Galuszka et al., 2007*; *Kudo, Kiba & Sakakibara, 2010*; *Sakakibara, 2006*). The first step of iP and tZ biosynthesis is catalyzed by adenosine phosphate-isopentenyltransferases (IPTs). IPTs produce iP-ribotides from dimethylallyl diphosphate (DMAPP) and adenosine 5′-diphosphate (ADP) or adenosine 5′-triphosphate (ATP) (*Ihara et al., 1984*; *Taya, Tanaka & Nishimura, 1978*). iP-ribotides can then be hydroxylated to tZ-ribotides by cytochrome P450 monooxygenase, family 735, subfamily A (CYP735A) (*Takei, Yamaya & Sakakibara, 2004*). These cytokinin ribotides are converted to free-base CKs by cytokinin-activating enzymes LONELY GUYs (LOGs) (*Kurakawa et al., 2007*; *Kuroha et al., 2009*; *Tokunaga et al., 2012*). In addition, cZ and tZ can be enzymatically interconverted by zeatin *cis–trans* isomerase (*Bassil, Mok & Mok, 1993*; *Sakakibara, 2006*). In *Arabidopsis*, IPT1 and IPT3–IPT8 are involved in iP and tZ biosynthesis (*Kakimoto, 2001*; *Sun et al., 2003*; *Takei, Sakakibara & Sugiyama, 2001*), while IPT2 and IPT9 are involved in cZ biosynthesis (*Golovko et al., 2002*). *CYP735A1* is abundant in roots and flowers in *Arabidopsis*, while *CYP735A2* specifically accumulates in roots (*Takei, Yamaya & Sakakibara, 2004*). *CYP735As* are required for shoot growth (*Kiba et al., 2013*). Cytokinin oxidase/dehydrogenase (CKX) catalyzes the irreversible degradation of CKs (*Galuszka et al., 2001*, *2007*; *Schmulling et al., 2003*). *CKXs* play important roles in controlling CK levels in plant tissues. In *Arabidopsis*, *CKX3* and *CKX5* regulate the activity of reproductive meristems (*Bartrina et al., 2011*). In rice, *OsCKX4* mediates crown root development by integrating cytokinin and auxin signaling (*Gao et al., 2014*).

Cytokinins play important roles in flower bud development and floral sex differentiation (*Chandler, 2011*; *Gerashchenkov & Rozhnova, 2013*; *Yamasaki, Fujii & Takahashi, 2005*).

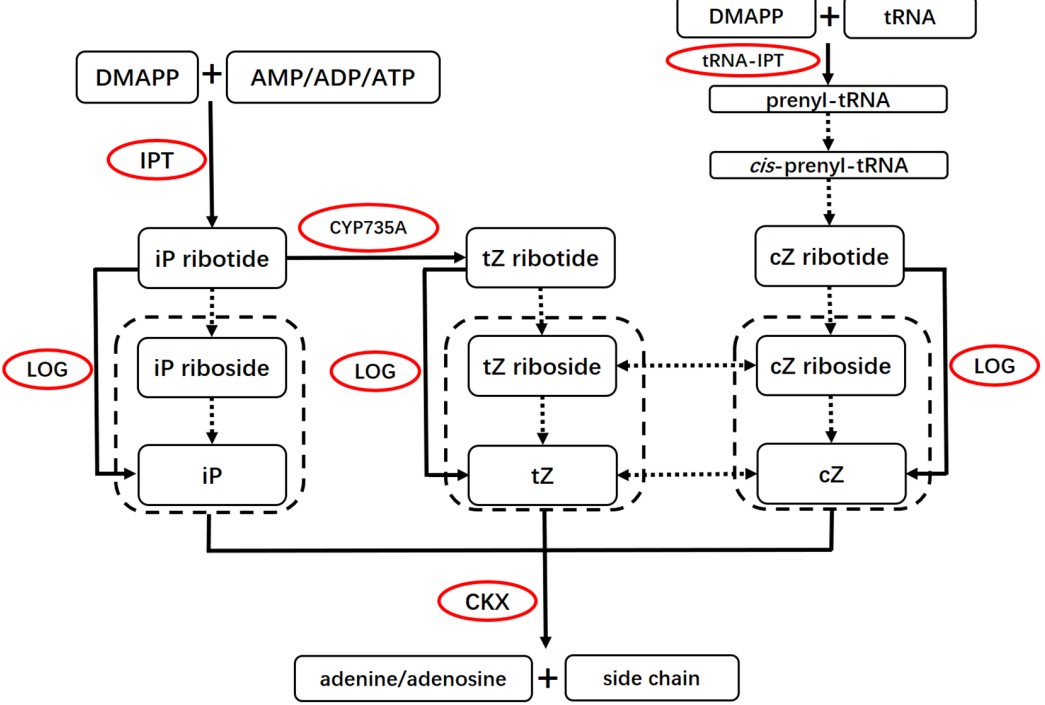

**Figure 1 Basic scheme for the cytokinin biosynthesis and degradation pathways.** Solid arrows indicate pathways with genes that are known, and dotted arrows indicate pathways with genes that remain to be identified. The enzymes are marked by red frames. The iP, *Z* and their ribosides inside the dotted boxes could be degraded by CKX. cZ, *cis*-zeatin; DMAPP, dimethylallyldiphosphate; CKX, cytokinin oxidase/dehydrogenase; cZ, *cis*-zeatin; DMAPP, dimethylallyl diphosphate; iP, N⁶-(Δ²-isopentenyl) adenine; IPT, adenosine phosphate-isopentenyltransferase; LOG, LONELY GUY; tRNA-IPT, tRNA-isopentenyltransferase; tZ, *trans*-zeatin; Z, zeatin. This figure was modified and redrawn from reference (*Kudo, Kiba & Sakakibara, 2010*).

However, the roles of CK biosynthesis genes *IPTs* and *CYP735A* and catabolism gene *CKXs* in *Jatropha* are not clear. In this study, we isolated sequences of cytokinin metabolic genes, including six *IPTs*, one *JcCYP735A*, and seven *JcCKXs*, using the *Jatropha* Genome Database (*Hirakawa et al., 2012*; *Sato et al., 2010*; *Wu et al., 2015*). The 14 genes showed different expression patterns in different tissues of *Jatropha*. Some of them exhibited tissue-specific expression. *JcIPT6* was only expressed in seeds a few days after pollination. *JcCYP735A* was highly expressed in roots and seeds. *JcCKX4* was expressed mainly in seeds. In addition, CK types and contents were detected in flower buds and seeds. With flower bud development iP-type CKs increased, while tZ-type CKs decreased. With seed development, tZ-type CKs increased, while iP-type CKs decreased. The *Jccyp735a* mutants were obtained by the clustered regularly interspaced short palindromic repeats (CRISPR)-Cas9 system. Compared with the wild-type (WT) plants, the concentrations of tZ and tZ-riboside (tZR) decreased significantly in the *Jccyp735a* mutants, which showed severely retarded growth. These results will be helpful for future studies of the functions of these genes and for improving the biological characteristics of *Jatropha*.

## MATERIALS AND METHODS

### Plant materials and growth conditions

Three-year-old *Jatropha* trees were grown in the field at Xishuangbanna Tropical Botanical Garden of the Chinese Academy of Sciences, Mengla County, Yunnan Province, China (21°54′N, 101°46′E; 580 m in altitude). The seedlings of WT and the $T_1$ plants of *Jccyp735a* mutants were grown in the greenhouse (28°C, 12 h light/12 h dark, 70% humidity). Flower buds, ovules, and seeds at different developmental stages were collected in May–July 2015 for quantitative reverse transcriptase-polymerase chain reaction (qRT-PCR) analysis of *JcIPT6* expression and quantification of cytokinin contents. All other samples used in qRT-PCR expriments were collected at the same time in May 2015. Various plant tissue samples, including lateral roots of 1–2 mm in diameter with fine-roots and root tips, shoot apex of 0.3 cm in length from the top of shoots, stems of 1.5 cm in diameter, young leaf blades of 2 cm in length, mature leaf blades of 15 cm in length, flower buds of 0.3 cm in length, just-opened female and male flowers, fruits of 15 days after pollination, and seeds of 30 days after pollination, were harvested for qRT-PCR analysis. All tissues were immediately frozen in liquid nitrogen and stored at −80 °C until needed.

### Gene identification and isolation

Sequences of orthologous *IPT*, *CYP735A*, and *CKX* genes from *Arabidopsis* that were available in the GenBank database were used as query sequences for basic local alignment search tool (BLAST) analysis using GenBank, the *Jatropha* Genome Database (http://www.kazusa.or.jp/jatropha/index.html) and our *Jatropha* transcriptome data (*Chen et al., 2014*; *Pan et al., 2014*). The full length of complementary DNA (cDNA) and genomic DNA sequences of *JcIPT*, *JcCYP735A*, and *JcCKX* were obtained by PCR amplification. The PCR products were subsequently cloned into the pGEM-T vector (Promega Corporation, Madison, WI, USA) and sequenced. The GenBank accession numbers for the nucleotide sequences of these genes are listed in Table S1. Primers used in PCR are listed in Table S2.

### Sequence comparison and phylogenetic analysis

Sequence chromatograms were examined and edited using Chromas Version 2.23 (http://technelysium.com.au/). Related sequences were identified with BLAST (http://www.ncbi.nlm.nih.gov/BLAST/). Genomic organization of all genes was analyzed by using the Gene Structure Display Server (GSDS) with default settings (*Hu et al., 2015*). A phylogenetic tree was generated with MEGA 7.0 (http://www.megasoftware.net/) using the Poisson model with gamma-distributed rates and 1,000 bootstrap replicates.

### Expression pattern analysis by qRT-PCR

Total RNA was extracted from each tissue, and first-strand cDNA was synthesized with a PrimeScript® RT Reagent Kit with gDNA Eraser (Takara, Dalian, China) according to the manufacturer's instructions. qRT-PCR was performed with LightCycler® 480 SYBR Green I Master (Roche, Indianapolis, IN, USA) on the Roche 480 Real-Time PCR Detection System (Roche Diagnostics, Mannheim, Germany). qRT-PCR was performed

with two independent biological replicates (tissue samples were harvested from different plants) and three technical replicates for each sample. Data were analyzed using the $2^{-\Delta\Delta CT}$ method as described by *Livak & Schmittgen (2001)*. Expression levels of specific genes were normalized to that of the *actin* gene in *Jatropha* (*Zhang et al., 2013*). Primers used in qRT-PCR are listed in Table S3.

## Quantification of cytokinin

Cytokinin contents were determined by the Wuhan Greensword Creation Technology Co. Ltd., using a polymer monolith microextraction coupled with hydrophilic interaction chromatography-tandem mass spectrometry method as described previously (*Liu, Wei & Feng, 2010*).

The leaves used to quantify the CKs were the third and fourth new leaves from four-month-old WT and the $T_1$ plants of *Jccyp735a* mutants. Three independent biological replicates and three technical replicates were measured for each sample. The data were analyzed using the Statistical Product and Service Solutions software (SPSS Inc., Chicago, IL, USA, version 16.0). Differences among the means were determined using a one-way ANOVA with Tukey's or Tamhane's post hoc tests ($p < 0.05$).

## Construction of CRISPR/Cas9 vectors and transformation of *Jatropha*

The sequence of *JcCYP735A* (GenBank accession no. XM_012222581.2) was analyzed with the online tool CRISPR-P (http://cbi.hzau.edu.cn/crispr/) to find the target sites of CRISPR/Cas9. pYLsgRNA-AtU3d/LacZ (GenBank accession no. KR029100) as the single-guide RNA (sgRNA) intermediate plasmid, and pYLCRISPR/Cas9P$_{35S}$-N (GenBank accession no. KR029112) as the binary vector were used for the CRISPR-Cas9 construction of *JcCYP735A* following the instruction of the CRISPR-Cas9 system (*Ma et al., 2015*). Transformation of *Jatropha* with *Agrobacterium* strain EHA105 carrying the *JcCYP735A* CRISPR/Cas9 construction was performed according to the protocol described by *Fu et al. (2015)*. The *Jccyp735a* mutants in transgenic *Jatropha* plants were identified by PCR amplification and DNA sequencing using a pair of primers, XB619 (5′-ATGGCCATGATATTAACAACTCTATTAG-3′) and XB620 (5′-GCGGTTCTATCCCATTCCAGTATAT-3′).

## RESULTS

### Cloning and identification of *JcIPTs*, *JcCYP735A*, and *JcCKXs*

Using all annotated *Arabidopsis* IPT, CYP735A, and CKX family members in the TAIR as query sequences to perform a BLAST analysis in GenBank and with our *Jatropha* transcriptome data (*Chen et al., 2014*; *Pan et al., 2014*), we identified and cloned *IPT*, *CYP735A*, and *CKX* orthologous sequences in *Jatropha*. The *Jatropha* IPT family included only six members, while there are nine members in *Arabidopsis*. These genes were named *JcIPT1*, *JcIPT2*, *JcIPT3*, *JcIPT5*, *JcIPT6*, and *JcIPT9*. The BLAST analysis identified only one member of the *CYP735A* family, *JcCYP735A*. The *CKX* gene family, encoding

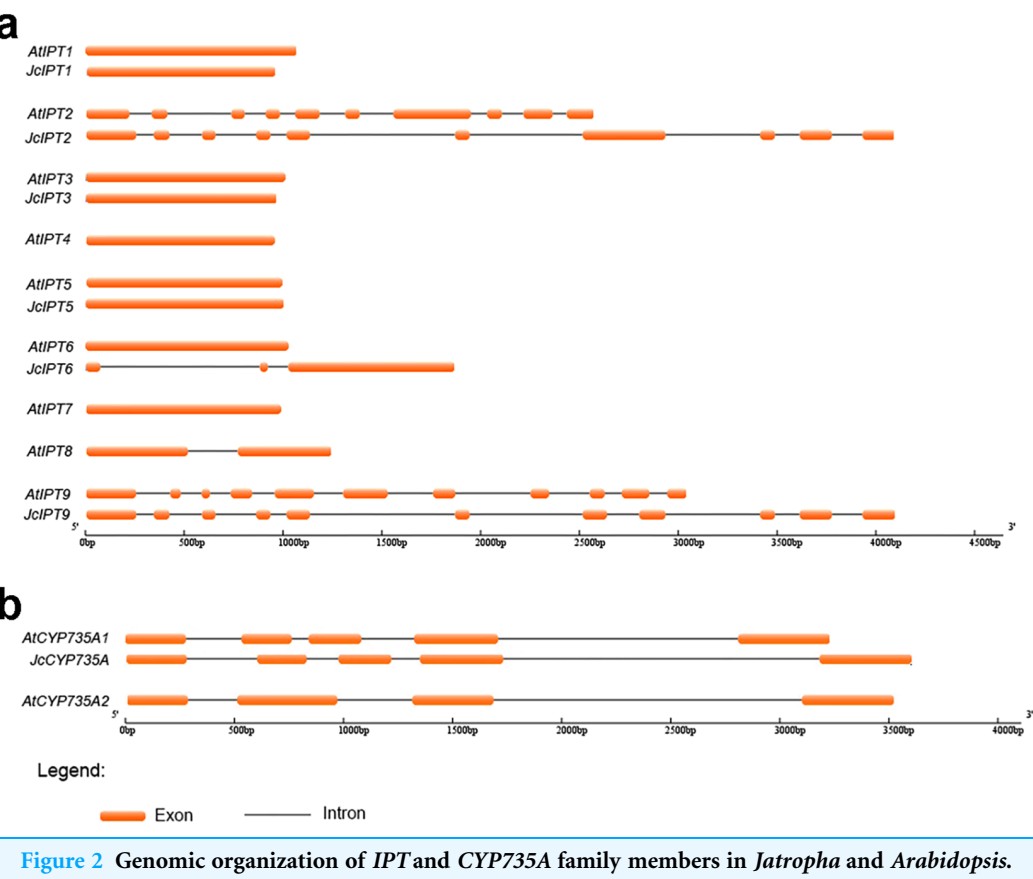

**Figure 2 Genomic organization of *IPT* and *CYP735A* family members in *Jatropha* and *Arabidopsis*.** (A) *IPT* family members; (B) *CYP735A* family members. At, *Arabidopsis thaliana*; Jc, *Jatropha curcas*.

degradation enzymes, included the same seven members in *Jatropha* as in *Arabidopsis*. These genes were named *JcCKX1*, *JcCKX2*, *JcCKX3*, *JcCKX4*, *JcCKX5*, *JcCKX6*, and *JcCKX7*.

Sequence structure analysis showed that *IPT*, *CYP735A*, and *CKX* family members shared almost the same numbers of exons and introns between *Jatropha* and *Arabidopsis* and had similar exon lengths (Figs. 2 and 3). *JcIPT6* has two more exons than *IPT6* from *Arabidopsis*. However, the extra two exon sequences are short and are not part of the P-loop NTPase domain (Fig. 2A).

## Phylogenetic analysis of JcIPTs, JcCYP735A, and JcCKXs

To analyse the phylogenetic relationships between orthologous genes, phylogenetic analysis were performed. IPT, CYP735A, and CKX family members from *Arabidopsis thaliana*, *Ricinus communis*, and *Oryza sativa* were compared with those from *Jatropha*. Orthologues of IPT1, 2, 3, 5, and 7 formed a clade, while IPT9 formed a single clade (Fig. 4A). JcCYP735A along with other dicotyledon CYP735As formed a clade, while CYP735A3 and 4 of *O. sativa* formed another clade (Fig. 4B). Orthologues of CKX1, 5, 6, and 7 formed a clade, while those of CKX2, 3, and 4 formed a separate clade (Fig. 5). These results showed that JcIPTs, JcCYP735A, and JcCKXs were most closely related to genes from *R. communis*, which also belongs to the Euphorbiaceae family.

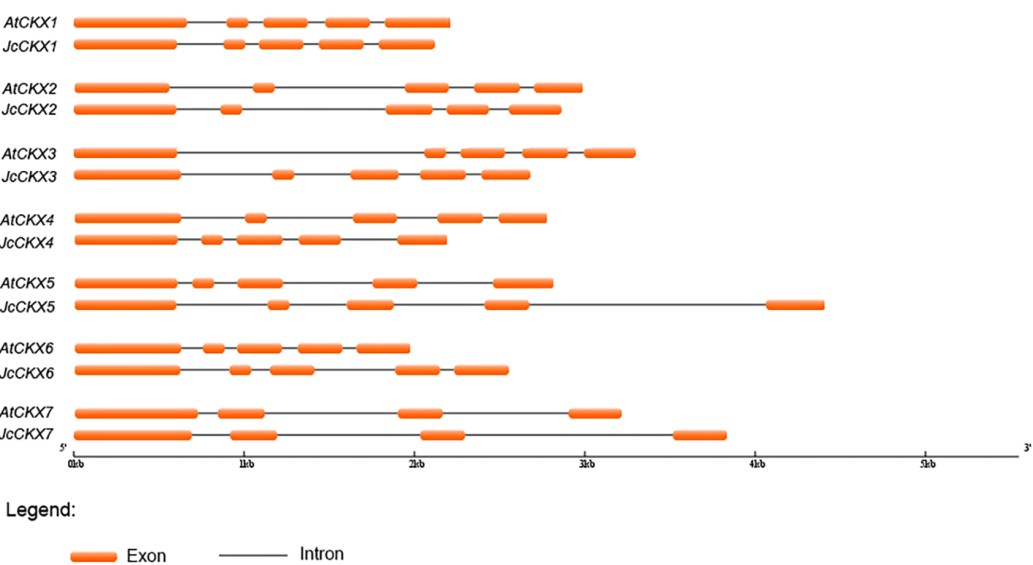

**Figure 3** **Genomic organization of *CKX* family members in *Jatropha* and *Arabidopsis*.** At, *Arabidopsis thaliana*; Jc, *Jatropha curcas*.         

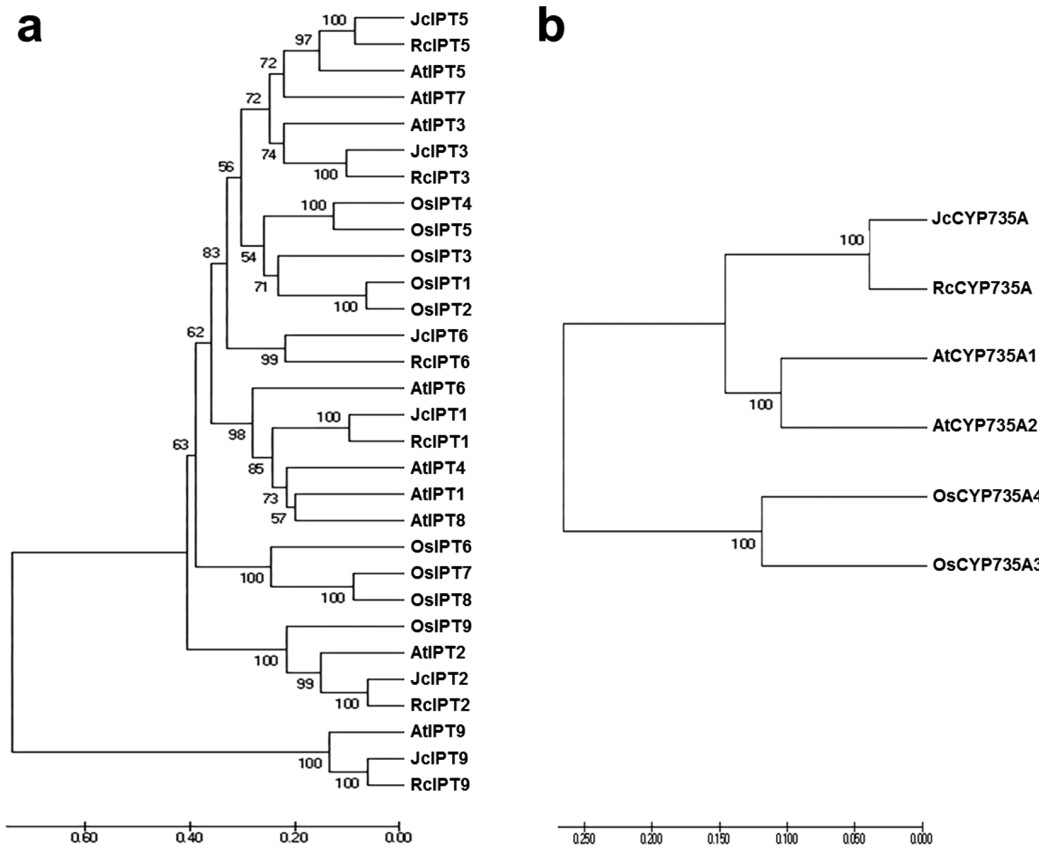

**Figure 4** **Neighbor-joining phylogenetic tree for IPT and CYP735A family members in various species.** (A) IPT family members; (B) CYP735A family members. At, *Arabidopsis thaliana*; Jc, *Jatropha curcas*; Os, *Oryza sativa*; Rc, *Ricinus communis*. 

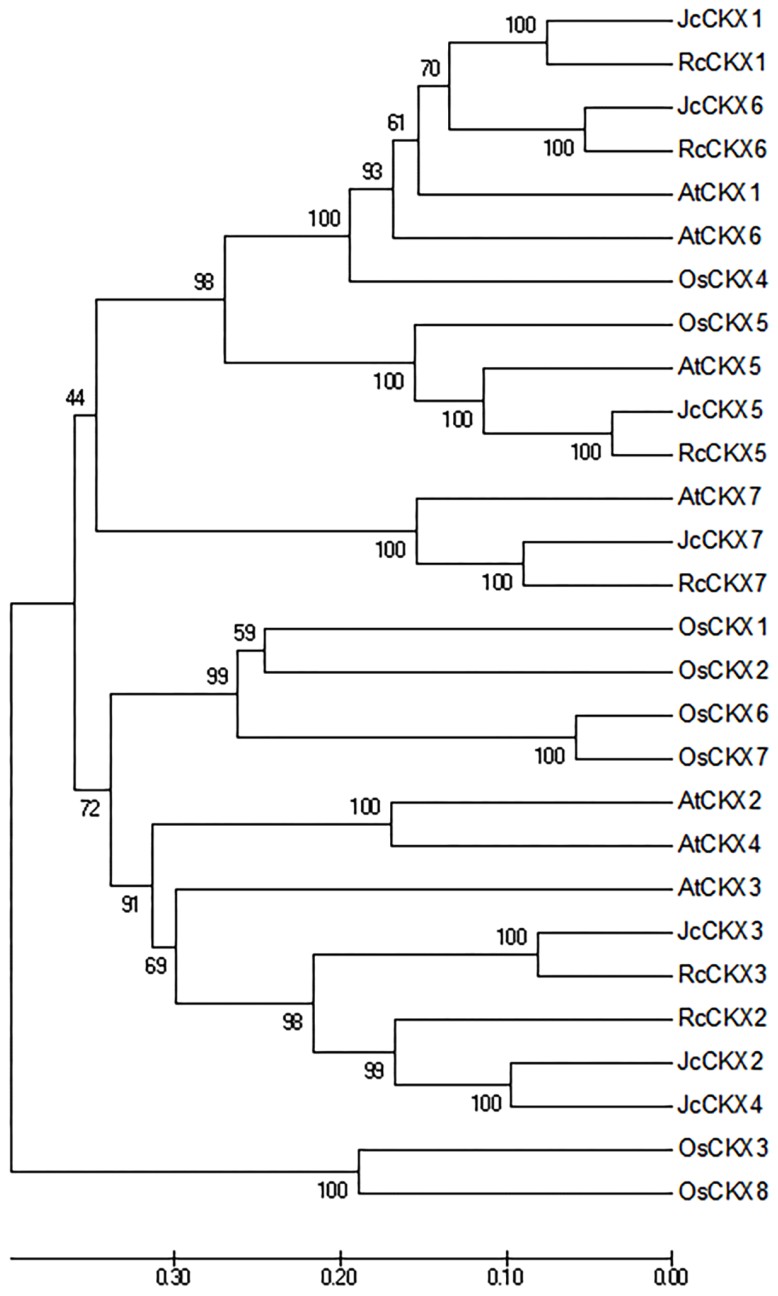

**Figure 5 Neighbor-joining phylogenetic tree for CKX family members in various species.** At, *Arabidopsis thaliana*; Jc, *Jatropha curcas*; Os, *Oryza sativa*; Rc, *Ricinus communis*.

## Expression patterns of *JcIPTs*, *JcCYP735A*, and *JcCKXs* in different tissues

In order to gain more information of these gene family members in *Jatropha*, the temporal and spatial expression patterns of these genes were analyzed using qRT-PCR. *JcIPT1* was mainly expressed in roots (Fig. 6A). *JcIPT2* was mainly expressed in roots, shoot

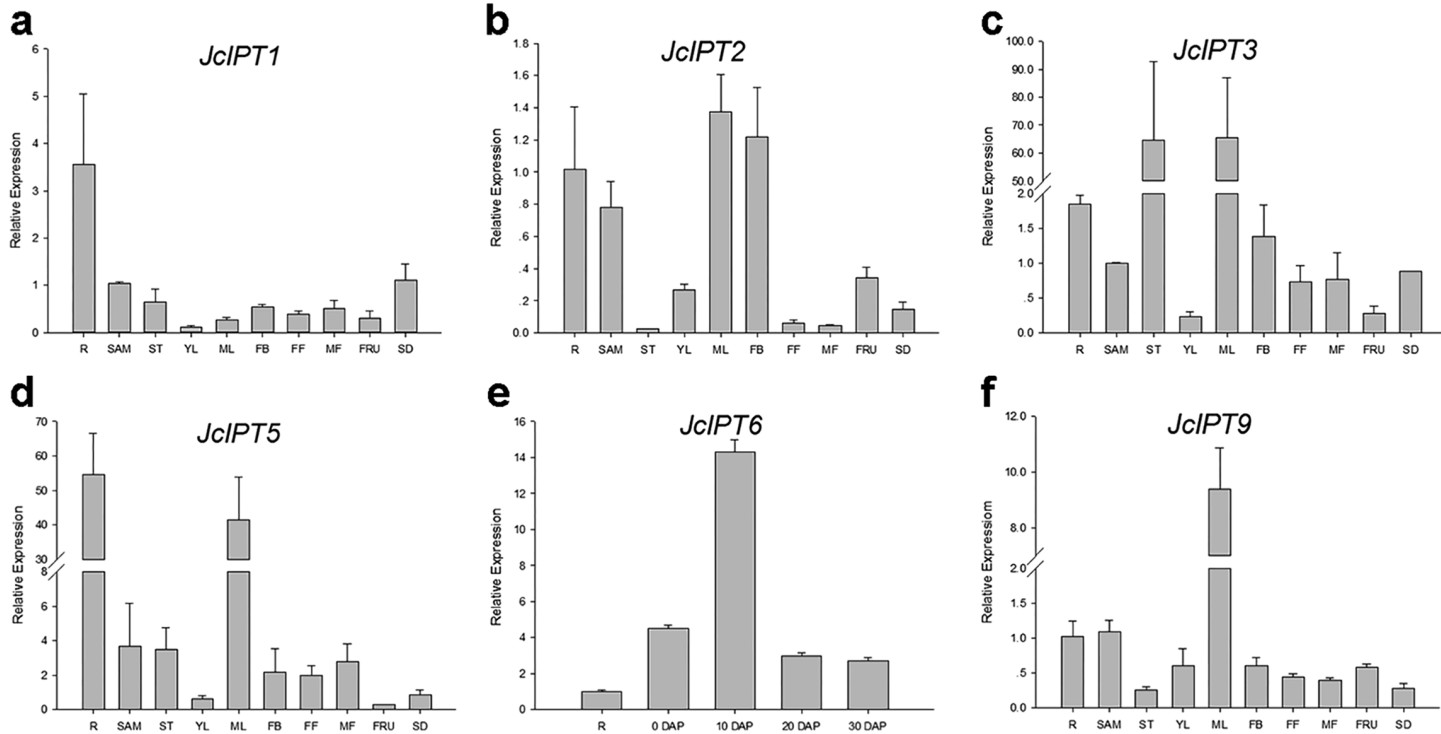

**Figure 6 Expression of *JcIPTs* in various *Jatropha* tissues.** (A–F) are expression patterns of *JcIPT1*, *JcIPT2*, *JcIPT3*, *JcIPT5*, *JcIPT6*, and *JcIPT9*, respectively. The qRT-PCR results were obtained from two independent biological replicates and three technical replicates for each sample. R, roots; SAM, shoot apical meristems; ST, stems; YL, young leaves; ML, mature leaves; FB, flower buds; FF, female flowers; MF, male flowers; FRU, fruits; SD, seeds; 0 DAP, unfertilized ovules; 10 DAP, seeds at 10 days after pollination; 20 DAP, seeds at 20 days after pollination; 30 DAP, seeds at 30 days after pollination; DAP, days after pollination.

apical meristems, mature leaves, and flower buds (Fig. 6B). *JcIPT3* showed much higher expression levels in stems and mature leaves than other tissues (Fig. 6C). *JcIPT5* exhibited high expression levels in roots and mature leaves (Fig. 6D). *JcIPT9* only showed high expression levels in mature leaves (Fig. 6F). The expression of *JcIPT6* was not detected in most of the plant tissues indicated above. After analysing more tissues (Fig. S1), we found that *JcIPT6* began to be expressed in seeds a few days after fertilization, with the strongest expression observed in seeds at 10 days after fertilization; the expression levels then decreased rapidly. In seeds at 20 days after fertilization, *JcIPT6* expression decreased by a factor of 5 compared with seeds at 10 days after fertilization (Fig. 6E). The expression levels of *JcCYP735A* were higher in roots, flower buds, and seeds than other tissues (Fig. 7A). *JcCKX1* was mainly expressed in flower buds, roots, and female flowers (Fig. 7B). *JcCKX2* showed very strong expression in female flowers and seeds (Fig. 7C). *JcCKX3* was highly expressed in male flowers (Fig. 7D). *JcCKX4* exhibited high expression levels in mature leaves and female flowers, and extremely high expression in seeds (Fig. 7E). *JcCKX5* was mainly expressed in stems, young leaves, and fruit (Fig. 7F). *JcCKX6* was expressed in all tissues (Fig. 7G). *JcCKX7* was mainly expressed in roots (Fig. 7H).

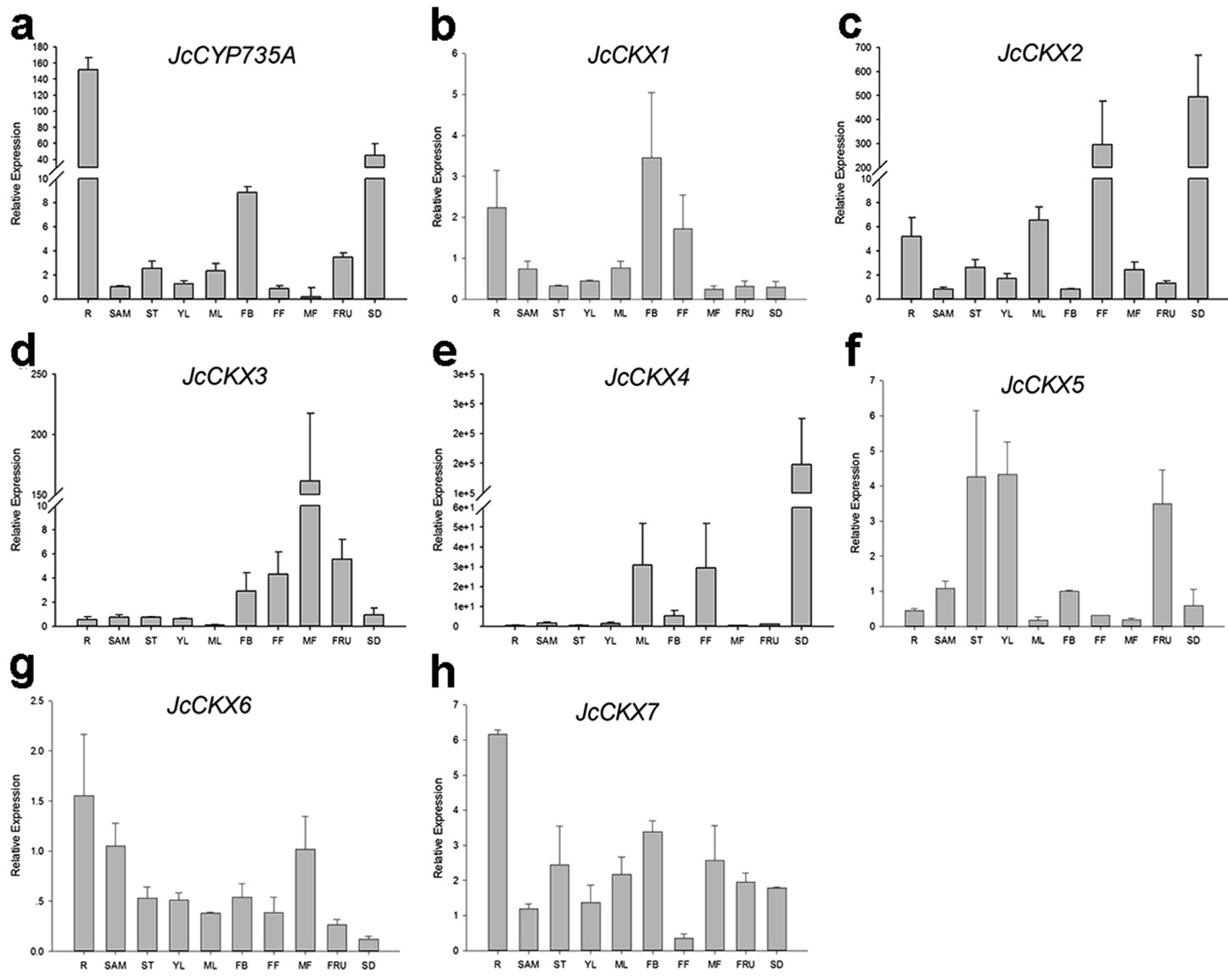

**Figure 7 Expression of *JcCYP735A* and *JcCKXs* in various *Jatropha* tissues.** (A–H) are expression patterns of *JcCYP735A, JcCKX1, JcCKX2, JcCKX3, JcCKX4, JcCKX5, JcCKX6,* and *JcCKX7,* respectively. Values in the *y*-axis of (E) are displayed in scientific notation. e, exponent. The qRT-PCR results were obtained from two independent biological replicates and three technical replicates for each sample. R, roots; SAM, shoot apical meristems; ST, stems; YL, young leaves; ML, mature leaves; FB, flower buds; FF, female flowers; MF, male flowers; FRU, fruits; SD, seeds.

### Endogenous CK contents in flower buds and seeds

In order to learn more about the distribution of endogenous cytokinins, we measured the contents of endogenous CKs in *Jatropha* flower buds and seeds at different developmental stages (Fig. S1). Different profiles were observed for each CK variant with flower bud and seed development in *Jatropha* (Table 1). The contents of iP and its variant iP-riboside (iPR) increased approximately 177-fold and nine-fold, respectively, from the flower bud 1 (FB1) stage to the flower bud 2 (FB2) stage. Compared with the FB1 stage, tZ content was approximately doubled in the FB2 stage, while tZR content
**Table 1 Content of endogenous CKs in flower buds and seeds of *Jatropha* in different developmental stages (ng/gFW).**

| Samples | iP | iPR | iP9G | tZ | tZR | tZ9G | DZ | DZR |
|---------|-----|------|------|-----|------|------|-----|------|
| FB1 | 0.14 ± 0.01 | 1.50 ± 0.03 | N.D. | 1.21 ± 0.03 | 12.55 ± 0.78 | N.Q. | 1.27 ± 0.09 | 9.24 ± 0.15 |
| FB2 | 24.86 ± 0.83 | 15.24 ± 0.24 | N.D. | 2.49 ± 0.17 | 0.89 ± 0.08 | N.D. | 3.31 ± 0.19 | 0.97 ± 0.03 |
| 0 DAP | 3.97 ± 0.40 | 1.20 ± 0.07 | N.D. | 0.79 ± 0.08 | 0.08 ± 0.007 | N.D. | 0.5 ± 0.06 | N.D. |
| 10 DAP | 3.96 ± 0.19 | 0.71 ± 0.02 | N.D. | 22.61 ± 2.42 | 1.56 ± 0.09 | 0.06 ± 0.001 | 6.42 ± 0.24 | 0.18 ± 0.02 |
| 20 DAP | 0.31 ± 0.01 | 0.78 ± 0.05 | N.D. | 149.28 ± 11.60 | 39.76 ± 1.80 | 0.75 ± 0.03 | 28.90 ± 2.68 | 16.56 ± 0.86 |

Notes:
FB1, flower buds of less than 5 mm in length; FB2, flower buds of two to three cm in length; 0 DAP, unfertilized ovule; 10 DAP, seeds of 10 days after pollination; 20 DAP, seeds of 20 days after pollination; DAP, days after pollination; N.D., not detected; N.Q., not quantified.

was reduced by 93%, resulting in a decrease in the amount of total active tZ variants. Conversely, the contents of tZ variants increased remarkably during seed development; compared with ovules, the tZ content increased 187-fold and the tZR content increased 496-fold in seeds at 20 DAP.

### *Jccyp735a* mutants generated by CRISPR-Cas9 system showed retarded growth

To explore the biological function of *JcCYP735A* in *Jatropha*, we generated *Jatropha* transformants with *JcCYP735A* knocked out using the CRISPR-Cas9 system (Fig. 8). Three homozygous mutant lines, L1, L2, and L3, were obtained by DNA sequencing (Fig. 8A). Endogenous contents of CKs in the leaves of two lines of *Jccyp735a* mutants (L2 and L3) and WT plants were examined. The results showed that the concentrations of tZ and tZR, and cZ and cZR significantly decreased, whereas the concentrations of iP and iPR significantly increased in *Jccyp735a* mutants compared with those of the WT (Fig. 8B). *Jccyp735a* mutants showed severely retarded growth, and the mutant plants were only approximately a quarter the height of the WT plants (Figs. 8C and 8D).

### DISCUSSION

In our study, six *IPT* family members were identified in *Jatropha*. The number of *IPT* genes differs among plant species; for example, there are six *IPTs* in *R. communis*, while there are nine *IPTs* in both *Arabidopsis* (Kakimoto, 2001; Takei, Sakakibara & Sugiyama, 2001) and rice (Fig. 4A). *JcIPTs* have the same number of exons as those in *Arabidopsis*, except *JcIPT6*, which has two more exons than *AtIPT6*. The third exon of *JcIPT6* has almost the same number of base pairs as the only exon in *AtIPT6* (Fig. 2A). It appears that the other two small exons were lost during evolution. Expression pattern analysis revealed that different *JcIPT* members were expressed in different tissues in *Jatropha* (Fig. 6). *JcIPT1* had an expression pattern similar to that of *AtIPT1* and was mostly expressed in roots, shoot apical meristems, and seeds (Miyawaki, Matsumoto-Kitano & Kakimoto, 2004). JcIPT2 and JcIPT9 were assigned to the same cluster as their orthologues (Fig. 4A). In *Arabidopsis, AtIPT2* and *AtIPT9* are expressed ubiquitously, with stronger expression in proliferating tissues, including the root and shoot apical meristems and leaf primordia (Miyawaki, Matsumoto-Kitano & Kakimoto, 2004). Similarly, *JcIPT2* and *JcIPT9* were expressed ubiquitously in *Jatropha*. However, the strongest expression of both was in

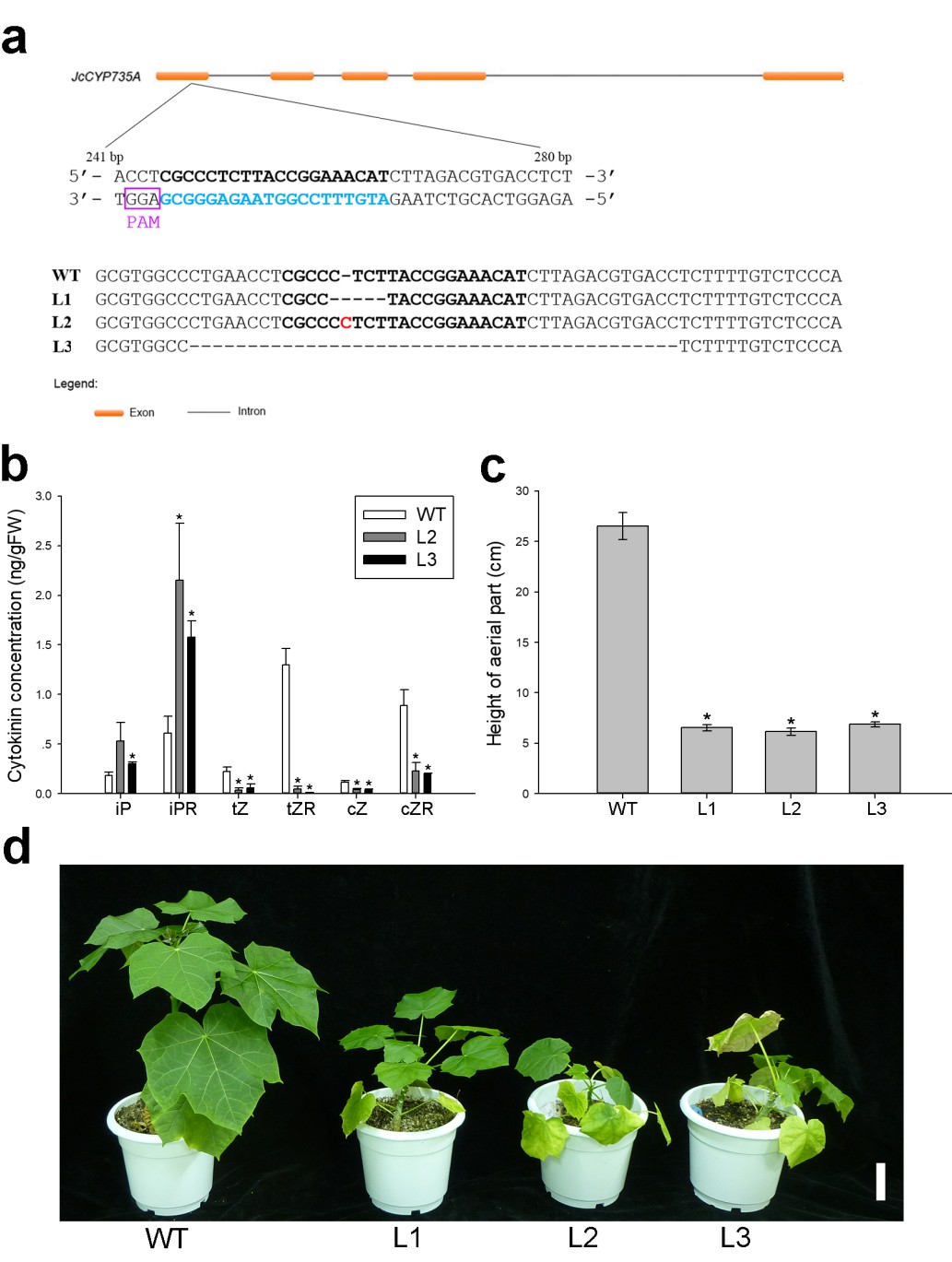

**Figure 8 Generation and phenotypic variation of *Jccyp735a* mutants of *Jatropha* obtained by the CRISPR-Cas9 system.** (A) Different types of *JcCYP735A* mutation generated by the CRISPR-Cas9-mediated gene silencing in the progenies ($T_1$ generation) of the three transgenic lines L1, L2, and L3. The blue characters show the selected target site sequences. The characters in purple box show the proto-spacer adjacent motif (PAM) sequences. The red character indicates the nucleotide insertion. WT, wild-type. (B–D) Cytokinin concentrations (B), height (C), and appearance (D) of the four-month-old seedlings ($T_1$) of *Jccyp735a* mutant lines and WT. Error bars represent the standard deviation (SD) of three (B) or five (C) biological replicates. Asterisks indicate statistically significant differences compared with WT ($p < 0.05$). In (D), bar = 5 cm. Photo by Li Cai.

mature leaves instead of meristems and young leaves (Figs. 6B and 6F). Considering that AtIPT2 and AtIPT9 are tRNA-IPTs, which only catalyze cZ biosynthesis (*Golovko et al., 2002*), cZ might be produced mainly in mature leaves of *Jatropha*. AtIPT3 was found to be responsible for nitrate-dependent cytokinin biosynthesis and is predominantly expressed in the phloem (*Miyawaki, Matsumoto-Kitano & Kakimoto, 2004*; *Takei et al., 2002*). In our study, *JcIPT3* was strongly expressed in stems and mature leaves, which contain abundant phloem (Fig. 6C). A study by *Miyawaki, Matsumoto-Kitano & Kakimoto (2004)* showed that GUS activity was not detected in *Arabidopsis* transformants carrying *AtIPT6::GUS*. RT-PCR analysis indicated that it was abundant in siliques. Analogously, we did not detect *JcIPT6* in most tissues of *Jatropha*, including fruits. We found that *JcIPT6* was mainly expressed in seeds at 10 days after pollination, and its expression decreased rapidly thereafter (Fig. 6E). It is possible that the previous study missed this critical phase in the seeds chosen for GUS staining (*Miyawaki, Matsumoto-Kitano & Kakimoto, 2004*). Our results suggested that different *JcIPT* family members play different roles in the development of *Jatropha* and that some *JcIPT* members could be used to cultivate high-yield varieties of *Jatropha* using transgenic technology.

Although none of the ATP/ADP *JcIPTs* were found to be highly expressed in flowers, iP, iPR, and tZ contents increased with the development of flower buds (Table 1). It is known that CKs can be transported through the plant vascular system (*Hirose et al., 2007*). *JcCYP735A* was found to be highly expressed in flower buds (Fig. 7A). iP-type CKs may be transported into flowers, and some of them may then be used to generate tZ-type CKs via JcCYP735A. In addition, it has been reported that the tZ concentration is up-regulated in the early development of tomato fruits (*Matsuo et al., 2012*). Our study showed that *JcCYP735A* was highly expressed and that the tZ concentration increased with seed development in *Jatropha* (Table 1). Thus, *JcCYP735A* might play important roles in seed development by controlling tZ biosynthesis. Similar to *R. communis* (*Chan et al., 2010*), only one *CYP735A* gene was found in *Jatropha*, although there are two *CYP735A* members in *Arabidopsis* and rice (*Takei, Yamaya & Sakakibara, 2004*; *Tsai et al., 2012*).

Unlike *JcIPT* or *JcCYP735A* family, the *JcCKX* family in *Jatropha* has the same number of members as that in *Arabidopsis* (Fig. 3). In addition, the *JcCKX* and *AtCKX* orthologues contain the same number of exons, with five exons in *CKX1-6* and 4 exons in *CKX7* (Fig. 3). Further expression analysis showed that *JcCKX2* was mostly expressed in female flowers, whereas *JcCKX3* was mostly expressed in male flowers (Figs. 7C and 7D). Both *JcCKX2* and *JcCKX4* were expressed strongly in seeds (Figs. 7C and 7E). These tissue-specific expression genes may be chosen to adjust the CK content in these tissues using transgenic methods. It has been reported that reduced expression of *OsCKX2* causes cytokinin accumulation in inflorescence meristems and increases the number of reproductive organs, resulting in enhanced grain yield (*Ashikari et al., 2005*). Decreased expression of *CKX* orthologues may also lead to increased yield in *Jatropha*. Moreover, overexpression of some *CKX* members can also improve resistance. Overexpression of *CKX1* or *CKX2* in *Arabidopsis* and other species causes elongation of the primary root and increases root branching (*Galuszka et al., 2004*; *Mrízová et al., 2013*; *Pospisilova et al., 2016*; *Werner et al., 2001*; *Yang et al., 2003*), while overexpression of *AtCKX7* results in an

opposite phenotype (*Kollmer et al., 2014*). Specific expression of *JcCKX1* or *JcCKX2* in roots might be used to transform a shallow root system into a deep root system to improve the growth and lodging resistance of *Jatropha*. Furthermore, root system development might enhance tolerance to drought stress. Remarkably, *JcCKX4* expression was much higher in seeds than other tissues (Fig. 7E), suggesting that JcCKX4 may be a key enzyme regulating cytokinin levels to affect seed development.

In early flower bud development, the content of iP-type CKs increased significantly while that of tZ-type CKs decreased. This result indicated that iP-type CKs participate more in early flower bud development than tZ-type CKs. In tomato, iP-type CK contents decrease during fruit ripening (*Matsuo et al., 2012*). By contrast, in early seed development, the content of tZ-type CKs increased substantially, while that of iP-type CKs decreased. This result, which is in accordance with high expression of *JcCYP735A* in seeds, suggested that tZ-type CKs are dominant in early seed development. Many differences were observed in CK contents in different periods of *Jatropha* flower bud and seed development. Our results indicate that iP-type CKs can be used to improve the number of flowers in *Jatropha*, while tZ-type CKs can be used to enlarge seeds.

The single *cyp735a1* or *cyp735a2* mutant *Arabidopsis* showed slight decreases in tZ and tZR concentration compared with that of WT, while the double mutants showed great decreases (*Kiba et al., 2013*). In addition, *cyp735a1 cyp735a2* double mutants exhibited retarded shoot growth (*Kiba et al., 2013*). Similarly, in this study, *Jccyp735a* mutants showed substantial decreases in tZ and tZR concentrations (Fig. 8B), which is consistent with that only a single member of *JcCYP735A* was found in *Jatropha* genome (*Wu et al., 2015*). We noticed, however, tZ and tZR did not completely disappear in *Jccyp735a* mutants (Fig. 8B), which may result from conversion of cZ and cZR. This notion is supported by the fact that the concentrations of cZ and cZR also significantly decreased in *Jccyp735a* mutants (Fig. 8B), and that *cis–trans* isomerase activity for interconversion between cZ-type and tZ-type CKs has been reported in several plant species (*Bassil, Mok & Mok, 1993*; *Kudo et al., 2012*; *Suttle & Banowetz, 2000*).

## CONCLUSION

In this study, we isolated the members of the *JcIPT*, *JcCYP735A*, and *JcCKX* gene families and analyzed their temporal and spatial expression patterns. Different family members exhibited different expression patterns. Different types of CKs seemed to influence the development of flower buds and seeds, respectively. The analysis of the *Jccyp735a* mutants revealed that *JcCYP735A* plays an important role in tZ biosynthesis in *Jatropha*. These results will be helpful for further function studies of cytokinin metabolic genes and improving agronomic characteristics of *Jatropha* by genetic engineering of cytokinin metabolism.

## ABBREVIATIONS

**BLAST**    basic local alignment search tool
**CK**    cytokinin
**CKX**    cytokinin oxidase/dehydrogenase
**CRISPR**    clustered regularly interspaced short palindromic repeats

| CYP735A | cytochrome P450 monooxygenase, family 735, subfamily A |
|---|---|
| cZ | *cis*-zeatin |
| DZ | dihydrozeatin |
| DZR | dihydrozeatin-riboside |
| iP | $N^6$-($\Delta^2$-isopentenyl)-adenine |
| iP9G | $N^6$-($\Delta^2$-isopentenyl)-adenine-9-glucoside |
| iPR | iP-riboside |
| IPT | adenosine phosphate-isopentenyltransferase |
| LOG | LONELY GUY |
| qRT-PCR | quantitative reverse transcriptase-polymerase chain reaction |
| tZ | *trans*-zeatin |
| tZ9G | *trans*-zeatin-9-glucoside |
| tZR | tZ-riboside. |

## ACKNOWLEDGEMENTS

We thank Prof. Yao-Guang Liu (South China Agricultural University, China) for providing vectors pYLCRISPR/Cas9P$_{35S}$-N and pYLsgRNA-AtU3d. The authors gratefully acknowledge the Central Laboratory of the Xishuangbanna Tropical Botanical Garden for providing research facilities.

### Funding

This work was supported by the National Natural Science Foundation of China (Nos. 31370595, 31300568, and 31670612) and the Program of Chinese Academy of Sciences (Nos. ZSZC-014 and 2017XTBG-T02). The funders had no role in study design, data collection and analysis, decision to publish, or preparation of the manuscript.

### Grant Disclosures

The following grant information was disclosed by the authors:
National Natural Science Foundation of China: 31370595, 31300568, and 31670612.
Program of Chinese Academy of Sciences: ZSZC-014 and 2017XTBG-T02.

### Competing Interests

The authors declare that they have no competing interests.

### Author Contributions

- Li Cai performed the experiments, analyzed the data, prepared figures and/or tables, authored or reviewed drafts of the paper, approved the final draft.
- Lu Zhang performed the experiments, analyzed the data, approved the final draft.
- Qiantang Fu analyzed the data, contributed reagents/materials/analysis tools, prepared figures and/or tables, authored or reviewed drafts of the paper, approved the final draft.

![PeerJ](PeerJ logo)

- Zeng-Fu Xu conceived and designed the experiments, analyzed the data, contributed reagents/materials/analysis tools, prepared figures and/or tables, authored or reviewed drafts of the paper, approved the final draft.

## Data Availability

The GenBank accession numbers of the gene sequences described in this study are listed in Table S1.

## Supplemental Information

Supplemental information for this article can be found online at http://dx.doi.org/10.7717/peerj.4812#supplemental-information.

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
