# Peer review of "Identification and expression analysis of cytokinin metabolic genes IPTs, CYP735A and CKXs in the biofuel plant Jatropha curcas"

_PeerJ, doi:10.7717/peerj.4812_

## Round 0.1 · original submission · Major Revisions

To my mind it is essential that your on the cyp735a mutant is extended in a revised version of your manuscript and at least the lack of tZ-type cytokinins in these mutants should be demonstrated.

·

Basic reporting

Please see below general comments to author

Experimental design

Please see below general comments to author

Validity of the findings

Please see below general comments to author

Additional comments

The manuscript entitled "Identification and expression analysis of cytokinin metabolic genes IPTs, CYP735A and CKXs in the biofuel plant Jatropha curcas" is written well. The experiments were executed well and the results described and interpreted properly. The presented work possess novelty and described unambiguously. Potential of this study is related to increase or influence the seed yield in Jatropha. Therefore, I recommend this manuscript as suitable for publication, provided the authors revise the manuscript thoroughly based on points are given below.


Some of my specific comments are as follows:
1. The statistical analysis performed in this work was not described in the manuscript. So authors should describe the statistical tools used in this study.
2. Authors didn’t explain the role of LOG gene (cytokinin phosphoribohydrolase ‘Lonely guy’) in cytokinin biosynthesis. It would good to explain their role in cytokinin biosynthesis. Since it also plays a role in the formation of iP, tZ and cZ.
3. In line no 224, Kaori Miyawaki et al. should be rewritten as Miyawaki et al. (2004).
4. In Figure no 6 & 7 genes should be written in italics.
5. The Y- axis in the Figure no:7e should be clarified and mentioned in the figure legends.
6. Why author did not include LOG genes for expression analysis. ?

Reviewer 2 ·

Basic reporting

Please see in general comments.

Experimental design

Please see in general comments.

Validity of the findings

Please see in general comments.

Additional comments

Cai et al report on a number of cytokinin metabolism genes in the potential biofuel plant Jatropha. The paper is generally well written, however the information content is fairly low. The sequence data reported can principally be found in public databases and the construction of exon/intron structures, phylogenetic trees etc. (Fig. 1 to Fig. 5) is essentially a matter of few mouse clicks.
There is some new information in Fig. 6 which represents gene expression data. However, totally unacceptable is the claim made in line 128: “To deduce the functions of these gene family members … expression pattern were analysed.” Expression pattern are in no way informative about function, this is basic knowledge of biology, the statement needs to be adapted (E.g. “In order to gain more information ….”).
New Fig. 1 is correct with the exception that not only the free bases are degraded by CKX as is indicated in the scheme. In fact, CKX enzymes accept different substrates including the ribosides, ribotides and glucosides (see e.g. Galuszka et al., 2007). Please adapt figure accordingly. In the legend: line 3, In the red frames these are proteins, not genes. cis- and trans should be written italic. The figure misses to show the cytokinin conjugates.
New data concerning knockouts of the only CYP735A gene have been included. This is potentially a valuable and novel contribution but the data are premature. Even if a complete phenotypic analysis of the knockout lines might not be presented, a reasonable level of solid data should be reached. I am opposed to include preliminary data in the abstract and the result section of a manuscript. Either these plants are knockouts of cyp725A or not and either they have the phenotype as shown or not. If this is the case these data should not be described as preliminary but like “First analysis of knockout mutants of CYP735A …”
It is unclear why these data confirm that a single JcCYP725A gene is present in Jatropha (as is claimed in the abstract, line 25/26). One would need to confirm that these knockout plants do not contain trans-zeatin-type cytokinin, which are formed by CYP735A activity. As hormone analysis is straightforward and feasible with a reasonable input these data should be added. This is mandatory to support the claim that is made in the text.
Furthermore, the CRISPR/Cas9 construct used to generate is not completely reported. The complete sequence should be presented in Material and methods.
Why the three lines show the phenotype with different expressivity although all three appear to harbor a knockout mutation of the target gene? Please explain.
Minor points.
Line 18, IPT and CYP725A are collectively named cytokinin synthetase, which is not appropriate (see later in the text). Please check and adapt.
Line 161, “To confirm the activity and distribution of endogenous CKs, we detected …” is not justified. It is rather: “In order to learn more about the distribution of endogenous cytokinins we measured …”
Authors constantly confound genes (italic) and proteins. All text need to be checked carefully and adapted.
It is unclear whether the phenotype of the cyp735a plants is due to accelerated leaf senescence, it could simply be earlier leaf senescence. The phenotype should be compared to the one shown by cyp735a mutants of Arabidopsis and the differences discussed.

External reviews were received for this submission. These reviews were used by the Editor when they made their decision, and can be downloaded below.

---

## Round 0.2 · accepted · Accept

Your manuscript was send again to the more critical reviewer and he acknowledges that you have answered his questions.

# Reviewer 2 ·

Basic reporting

no comments

Experimental design

no comments

Validity of the findings

no comments

Additional comments

The authors have answered my questions, I recommend acceptance.

External reviews were received for this submission. These reviews were used by the Editor when they made their decision, and can be downloaded below.